# Influence of Parental Health Locus of Control on Behavior, Self-Management and Metabolic Control, in Pediatric Patients with Type 1 Diabetes

**DOI:** 10.3390/jpm12101590

**Published:** 2022-09-27

**Authors:** Roberto Franceschi, Marta Canale, Enrico Maria Piras, Lucia Galvagni, Cinzia Vivori, Vittoria Cauvin, Massimo Soffiati, Evelina Maines

**Affiliations:** 1Pediatric Diabetology Unit, Pediatric Department, Santa Chiara Hospital of Trento, 38122 Trento, Italy; 2Center for Information and Communication Technology, eHealth Unit, Fondazione “Bruno Kessler”, 38123 Trento, Italy; 3Center for Religious Studies, Bruno Kessler Foundation, 38123 Trento, Italy; 4Hygiene and Public Prevention Department, Azienda Provinciale per i Servizi Sanitari, 38100 Trento, Italy

**Keywords:** locus of control, caregiver, type 1 diabetes, pediatric

## Abstract

Background: Precision medicine in type 1 diabetes (T1D) treatment considers context and environmental data to subclassify patients. Parental Health Locus of Control PHLOC) could influence behavior, self-management, and metabolic control of children with T1D. Methods: No. 135 pediatric patients with T1D (No. 57 with HbA1c ≤ 7.0%, “optimal” group, and No. 78 with >7.0%, “sub-optimal” group) were enrolled in the study. History, anthropometric and diabetes management data were collected, as well as caregiver’s data about socioeconomic status (SES). The PHLOC scale questionnaire and a semi-structured interview were administered. Results: Access to technology was lower in the “sub-optimal” group and, in particular, in the ethnic minority subgroup, only 8% used them. In the “sub-optimal” group ethnic minority status was higher (24%), the caregiver had a lower SES and showed lower internal HLOC. Conclusions: New care models have to be implemented to ensure equity in diabetes care and precision treatment, particularly for ethnic minority groups, because SES and external PHLOC are still an important barrier to “optimal” diabetes control. In the “sub-optimal” group, we have to implement strategies aimed at increasing self-efficacy, while in the “optimal” one, a personalised approach should be considered to facilitate the shifting of responsibilities within the family, avoiding psychological distress.

## 1. Introduction

Despite the spread of new insulins and diabetes technologies, it is challenging for patients with type 1 diabetes (T1D) to achieve a target haemoglobin A1c (HbA1c) of <53 mmol/mol (<7.0%) [1]. Precision medicine in diabetes treatment is tailored to the individual and considers context data, environmental data, and glucose patterns derived from continuous glucose monitoring (CGM) systems to subclassify patients with T1D. This approach can improve person-centred outcomes and reduce costs [2].

Focusing on context and environmental data, T1D requires patients and their families’ daily management, and in the pediatric population, mothers are usually the main caregiver, although some fathers can play an active role, sharing an equal daily load in the management of responsibilities [3].

In other diseases, such as leukemia, Health Locus of Control (HLOC) of mothers has been shown to influence the coping styles of sick children [4]. In adults with T1D, an orientation to internal control or by healthcare professionals, are associated with better glycemic control and satisfaction with treatment [5,6]. However, these findings have not been replicated in other studies that found no effect of HLOC on patients’ beliefs, health knowledge, diabetes behaviors, or self-management [7]. In children–adolescents with T1D, different types of diabetes camp experiences changed HLOC towards an internal control, and this effect lasted for at least one year [8]. In the outpatient setting, children with optimal diabetes control (HbA1c < 7.5%), compared to children in suboptimal control, showed more “controlling” family environments and scored higher on the “power from others” subscale, according to the belief that health is determined by the influence of parents and healthcare professionals [9]. In parents of T1D youths, stress-reducing interventions based on training and relaxation techniques did not change their parental HLOC (PHLOC) in one study [10], while internal control was increased in another study [11].

Therefore, given the contradictory results presented in the literature, the aim of this study is to evaluate HLOC in the primary caregiver of youths with T1D and its relationship to diabetes behavior, self-management, and metabolic control.

## 2. Subjects and Methods

### 2.1. Subjects

The recruitment of participants was conducted at the Pediatric Diabetology Outpatient Clinic of Trento. No. 225 patients with diabetes, aged 1–18 years were followed up in April 2022 [12], and participants (children–adolescents and their caregivers) were selected according to the following inclusion criteria:

- patients with T1D aged 8–18 years at the time of recruitment;

- diabetes duration > 1 year;

- current insulin regimen (multi-injection or insulin pump) for at least 2 months;

- children that have never been diagnosed with psychiatric or behavioral disorders;

- primary caregivers were never diagnosed as psychiatric patients and were not assessed at risk of parental stress during the annual interview with the psychologist.

The exclusion criteria were the following:

- not meeting the inclusion criteria;

- the presence of complications related to diabetes or major pathology.

Some patients were also enrolled in other studies at our clinic [12,13].

### 2.2. Ethics Committee

The current study was approved by the Institutional Review Board of “Azienda Provinciale per i Servizi Sanitari della Provincia Autonoma di Trento” (reference number A786). The study was conducted from April 2022 to July 2022 on the occasion of a quarterly follow-up visit. Written informed consent was obtained from each participant and parent/legal guardian, as applicable, prior to enrolment. 

### 2.3. Study Design

Patients enrolled in the study underwent the quarterly control visit; history, anthropometric, and diabetes management data collected, are reported in Table 1. Based on the HbA1c value, participants were placed in an “optimal glycemic control” group [HbA1c ≤ 53 mmol/mol (7%)] or in a “suboptimal glycemic control” group [HbA1c > 53 mmol/mol (7%)], according to the ISPAD glycemic control targets [1].

We asked the caregiver accompanying the patient, data about family structure, nationality, language, religion, school education level, health literacy, and working status; the Parental Health Locus of Control Scale (PHLCS) questionnaire [14,15] was administered to the parent who was considered as the main caregiver.

A semi-structured interview was also administered to the accompanying caregiver/s and to the patient, with the purpose of understanding through a narrative analysis, the division of roles between caregivers and the adolescent’s copying style. The level of involvement of kids/adolescents in the interview varied according to their age, being minimal with younger children and reached its peak with 15–17 years old patients.

### 2.4. Methods

(1) The PHLCS questionnaire is a validated and standardised questionnaire, available in Italian, for the self-assessment of parental locus of control in relation to the children’s state of health [14,15]. The tool consists of 28 items divided into 6 subscales (children, parents, healthcare professionals, mass media, fate, and God), with possible answers on a 6-point Likert scale, ranging from 1 = I totally disagree to 6 = I totally agree. The Italian version of the questionnaire was validated in 2009 [15].

(2) The youths’ coping styles were studied through semi-structured interviews conducted by researchers of the Bruno Kessler Foundation (FBK) with a background in social sciences and health humanities, with previous experience in qualitative research in healthcare in T1D (EP, LG). The narrative analysis approach has been proven effective in research conducted with populations with similar characteristics [16,17,18]. Individual or couple narrative interviews were conducted in a semi-structured manner following the topics reported in Appendix A; the topic guide represents a flexible guide to the interview. The semi-structured interviews were recorded and literally transcribed. The thematic analyses were conducted by adopting a template analysis [19], and a preliminary template was defined by the researchers (EP, LG) to investigate the sharing of burden of treatment between patients and caregivers and it was later adapted to identify recurring and emerging themes. The caregiver’s health literacy was assessed with the single item literacy screener (SILS), a question that identifies adults who need help in understanding information relating to written medical-health material [20].

(3) The family’s socioeconomic status (SES) was typified by collecting data regarding parents’ educational and occupational situations. Parents’ education level was classified as low (without a high school diploma: <14 schooling years) and high (high school diploma attainment or university studies: at least 14 schooling years) [21,22]. Parents’ occupation was collected following the classification of the Italian National Institute of Statistics, which is totally cross-linkable with the International Standard Classification of Occupations. They are grouped in two levels: low (unoccupied, unskilled, and semi-skilled workers, manual workers, and craftsmen), and high (legislators, senior officials, and managers, professionals, technicians, and associate professionals, sales workers, small business and farm owners, administrators and higher executives).

Family structure was investigated according to the following items: living with both parents (yes/no), the number of family members; ethnic minority status was defined as at least one parent born outside the country with a positive migration history, using a list of ethnic categories (African, Asian, mixed).

Areas of residence were classified as rural if they fall outside of settlements with more than 10,000 resident population.

Body mass index (BMI) was calculated as weight in kilograms divided by the square of height in metres. BMI-SDS was calculated using the WHO BMI charts [23].

Pubertal development was assessed according to Tanner staging [24]. Children–adolescents were classified according to three pubertal stages: pre-pubertal (equivalent to the Tanner stage 1), pubertal (Tanner stages 2 to 4), and post-pubertal (Tanner stage 5).

Capillary HbA1c level was measured using DCA Vantage^®^ Analyzer (Siemens Healthcare GmbH).

CGM data available in the 2-week period preceding every visit were collected.

### 2.5. Statistical Analysis

In order to determine the optimal number of patients to be consecutively enrolled in the study, when planning the present clinical trial, the calculation of the sample size was performed together with the statisticians. The primary outcome of the study was to identify differences in PHLC scores in caregivers of patients in “suboptimal glycemic control” compared to those in “optimal glycemic control”. We considered a minimum difference of 2.5 and a standard deviation of 4.2 as clinically relevant [5,6]. By accepting a two-tailed 5% α error and a study power of 90% (1-β), a numerosity (n) equal to 59 patients for each group is obtained. Statistics were analysed using GraphPad Prism version 8.0.2 (GraphPad, San Diego, CA, USA). Every dataset was tested for statistical normality and this information was used to apply the appropriate (parametric or nonparametric) statistical test. 

Data are expressed as mean ± SD for variables with normal distribution and with medians (interquartile range) for non-normally distributed variables. Differences between groups of continuous variables were analysed with t-student for paired samples for variables with normal distribution, or with Wilcoxon signed rank sum test for variables with non-normal distribution. The chi-square test with Fisher’s test has been used to evaluate differences in categorical data. Regression analysis was performed for parameters with *p* < 0.05.

The relationship between the PHLOC score and the patients’ metabolic control was evaluated and Spearman correlations were used to analyse statistical relationships between the different variables. Regression analysis was performed for parameters with *p* < 0.05. *p* values < 0.05 were considered significant.

## 3. Results

According to the inclusion/exclusion criteria, 152 patients were eligible for this study. Of these, 10 patients did not accept to participate and 7 were excluded because of incomplete data; therefore, 135 subjects were enrolled (68 males and 67 females) and No. 57 patients presented with HbA1c ≤ 7.0% (“optimal” group) and No. 78 with >7.0% (“sub-optimal” group). Among them, 30 subjects we randomly selected to be interviewed.

### 3.1. Demographic, Auxologic, and Metabolic Data

Complete data are reported in Table 1. The two groups were comparable in terms of gender, age at T1D onset, age at the study enrollment, diabetes duration, BMI z-score, pubertal stage, compliance to diabetology consultations, and associated diseases. Patients in the “optimal” group wore an insulin pump and real-time CGM more frequently in comparison to the “sub-optimal group” (*p* = 0.045 and 0.001, respectively). 

Time in range (TIR) was higher, while mean glucose and the coefficient of variation (CV) were lower in the “optimal” group (*p* = 0.002, <0.001, <0.001, respectively). Moreover, this group was more physically active and had lower insulin need (*p* = 0.001, *p* = 0.009 respectively). No patients experienced episodes of diabetic ketoacidosis in the year before the study enrollment, while 1 patient in the “sub-optimal group” had severe hypoglycemia.

Regression analysis in all the cohort, revealed that mean HbA1c in the last year was positively associated with HbA1c at the study enrollment, age at the study enrollment, total daily insulin dose, mean glucose values, CV, and negatively with percentage of time with active sensor and TIR (Appendix A).

### 3.2. Family and Caregiver Data

Complete data are reported in Table 2.

Considering patients with ethnic minority status (No. 25), only 6 (24%) were at HbA1c target and only 2 (8%) wear continuous subcutaneous insulin infusion (CSII) or real-time continuous glucose monitoring (rtCGM) systems. 

In the “sub-optimal” group, prevalence of at least one parent born outside the country was higher (24%, *p* = 0.040). The mother is more frequently the caregiver (72%), without differences between the two groups. Morocco caregivers were more represented in the “sub-optimal” group (*p* = 0.020). In the “optimal” group, the caregiver had a higher school education level (35% university, *p* < 0.001), health literacy and working status (*p* < 0.001). No differences in terms of the caregiver’s language or religion were detected.

Regression analysis in all the cohort, showed that HbA1c in the last year was negatively correlated with parents’ educational level and working status (Appendix A).

### 3.3. Parental Health Related Locus of Control Score

The results relative to the main caregiver locus of control obtained by PHLOC are shown in Table 3. Considering all the cohort, the caregivers reported an orientation to internal control or by healthcare professionals. Caregivers in the “sub-optimal” group showed lower internal HLOC (*p* = 0.001) and higher scores in God, fate and mass media scales (<0.001, <0.001, <0.001 respectively).

By correlation analyses, we investigated the association between caregivers’ PHLOC and the participants’ demographic, auxological and metabolic data. Internal PHLOC was positively associated with TIR and negatively with HbA1c in the last year. External PHLOC was associated with lower TIR, higher CV, lower education and occupation levels, ethnic minority status and religion other than Catholicism (Appendix A).

### 3.4. Youths’ Coping Styles

Semi-structured interviews were conducted on a representative sample of 30 families, 11 of the “optimal” group and 19 of the group “suboptimal”. 30 caregivers and 19 adolescents were interviewed. A detailed analysis of the interviews is out of the scope of the present article and will be presented in a separate paper. For the sake of this study we shall focus on the recurring and emerging themes and the coping styles.

Since the PHLOC was administred to caregivers, the joint interview of patients and family members allowed to investigate how the burden of responsibility is shared. The first result of our research is that shifting our focus from the caregiver to the family, internal locus of control of the caregiver may coexist with an external locus of control of children and adolescents as they rely on their caregivers.

A second emerging theme is the underlying (but sometimes explicit) tension between the caregiver coping style and the one desired by young patients, which expressed willingness to become more autonomous/acquire a larger autonomy, therefore to reduce the parental control, have more responsibilities, and have a more “relaxed” approach to self-management. This is consistent with the perceived trade-off between an ideal coping style and the limitations to social activities (e.g., staying over at a friend’s house). While qualitative and narrative data from the interviews cannot be mapped on the rigid categories of the PHLOC questionnaire, we may argue that teenagers, while having an internal locus of control may lean toward a fatalistic acceptance of long term consequences of adopting a less strict regime of adehernce to prescribed behaviours.

A third theme, strictly related to the second, is the dark side of the internal locus of control which, while being associated with better performances, comes at the cost of becoming “anxious” or “over scrupulous” as some of the parents defined themselves. As an example, some caregivers still perform night glucose checks waking up in the middle of the night and scanning the sensor while patients are asleep after 10 or more years. Such behavior is associated both with the fear that even some days off may hinder the health of their sons and daughters, on a short or longer term.

## 4. Discussion

We studied PHLOC in the main caregiver of youths with T1D, as contradictory results were available in the literature about HLOC in adults with T1D [5,6,7] and in parents of pediatric patients, tested above all during diabetes camp experiences [10,11]. Only one not-so-recent study reported data from an outpatient clinic and in total 56 children and their families were analysed [9]. According to us, these are the main findings of our study:(1)We confirm that mothers are frequently the main caregiver in the management in T1D (70–74%), as previously reported [16], without differences between “optimal” and “sub-optimal” groups. School education level, health literacy and working status of the main caregiver were lower in the “sub-optimal” group, compared to the “optimal” one, and in their families ethnic minority status was more prevalent. Access to technology (CSII or rtCGM) was lower in the “sub-optimal” group and, in particular, in the ethnic minority subgroup (No. 25), only 8% used them. The relationship between the caregiver’s low SES, ethnicity, and HbA1c was previously reported [25,26] and both independently influenced metabolic control [26]. Regarding access to technologies, lower use of CSII or rtCGM in T1D children from minority ethnic communities has been previously associated [27] to higher HbA1c; in Italy these patients have not to pay for diabetes technologies, therefore, we speculate that probably in ethnic minority groups, school education and health literacy are still an important barrier to access to diabetes technologies, and new care models as the “mosaic clinic” have to be implemented to ensure equity in diabetes care and precision treatment [28].(2)Parental locus of control was more internal (parental) in the “optimal group”, as previously reported in pediatric patients [9]. Instead, patients who failed to reach optimal metabolic control tended to show lower internal HLOC and higher scores in God, fate, and mass media. External PHLOC was associated with lower SES and ethnic minority status, and this data, according to a precision medicine approach, suggest the importance to assess PHLOC in patients with “sub-optimal” control, in order to implement strategies aimed at increasing self-efficacy. We suggest to take into account PHLOC and patients’ beliefs because this can positively impact medication adherence and minimise medication wastage, as previously reported [9]. Vice-versa, other authors did not find in adults with T1D any relationship between locus of control, health belief, knowledge, and diabetic self-management behavior or outcomes, and they suggested to tailor care for these patients to individual requirements [7].(3)Patients’ behavior and coping styles, explored by semi-structured interview, were demonstrated to be influenced by the caregivers’ PHLOC. This data is in agreement with the findings in children with leukemia that specific scales of PHLOC positively and negatively correlated with children’s coping style [4]. Our analysis, however, suggests that the “optimal group” would benefit from new strategies from healthcare professionals to facilitate the shifting of responsibilities within the family and addressing more directly the mental and psychological strain associated with the burden of treatment. This is consistent with the idea that T1D is a disease that affects the whole family, thus, caregivers should not only be empowered but also taken care of when needed.

As secondary findings of our study, we reported that in our centre, 48% of patients are at target (HbA1c ≤ 7%), compared to 28% with HbA1c < 7.5% reported in a recent survey in Italy [25]. We confirm the correlation between HbA1c and patients’ age, TIR and technology usage, as previously reported [25].

Strengths of this study are: (i) our study for the first time in literature, analysed PHLOC in the main caregiver of pediatric patients with T1D, in an outpatient clinic setting in a more numerous cohort than others reported in literature and we satisfied the power sample calculation; (ii) we used updated ISPAD target cut off for HbA1c (7%) to subclassify patients, and this parameter was suggested for epidemiological studies [1].

The limits of this study are: (i) it was conducted at a single site and other studies are needed to confirm our findings; (ii) we do not have measured C-peptide at the enrollment of this study to evaluate beta-cell mass in the two groups we have compared; (iii) we could not include time below range (TBR) < 5% as inclusion criteria for the “optimal control” because our patients wore different types of sensors; (iv) we evaluated SES collecting data regarding parents’ educational and occupational situation, while we did not have reliable data on economic income. 

In conclusion, in ethnic minority groups, school education, and health literacy are still an important barrier to coping with the disease, access technologies and achieve target HbA1c and healthcare professionals should take into account PHLOC in patients with “sub-optimal” control to implement strategies aimed at increasing self-efficacy [28]. Strategies aimed at increasing HLOC are awaited to improve their children’ coping strategies and to ensure improvement in metabolic control, according to a precision medicine treatment approach.

## Figures and Tables

**Table 1 jpm-12-01590-t001:** Population enrolled: demographic, auxological and metabolic data. 14 day time parameters of glucose control. Mean ± SD.

	All	HbA1c ≤ 7%	HbA1c > 7%	*p* Value
No. (%)	135	57 (42%)	78 (58%)	-
Male (%)/Female	68 (50%)/67	33 (58%)/24	35 (45%)/43	0.137
Age at T1D onset (years)	7.3 ± 3.9	8.1 ± 3.9	6.8 ± 3.8	0.08
Age at the study enrolment (years)	13.6 ± 3.0	13.6 ± 2.9	13.6 ± 3.1	0.89
Diabetes duration (years)	6.3 ± 3.9	5.5 ± 4.0	6.8 ± 3.8	0.06
Therapy:				
MDI	95 (70%)	32 (56%)	63 (81%)	0.002
CSII (SAP/HCL/AHCL)	40 (30%)	25 (44%)	15 (19%)	0.045
Glucose monitoring:				
Fingerstick (%)	7 (5%)	0	7 (9%)	-
isCGM (%)	94 (70%)	35 (61%)	59 (76%)	0.077
rtCGM (%)	34 (25%)	22 (39%)	11 (14%)	0.001
Total daily insulin (UI/Kg/die) (%basal)	0.82 ± 0.23 (54%)	0.76 ± 0.23 (54%)	0.87 ± 0.21 (54%)	0.009
BMI z-score	0.05 ± 0.98	−0.12 ± 0.94	0.17 ± 1.0	0.08
Prepubertal	35	14	21	0.75
Pubertal	56	22	34	0.56
Postpubertal	44	21	23	0.37
HbA1c at the study enrolment (%)	7.36 ± 1.2	6.54 ± 0.53	7.97 ± 1.20	<0.001
Mean HbA1c in the last year (%)	7.41 ± 1.12	6.52 ± 0.34	8.06 ± 1.03	<0.001
N. outpatient clinic visit/year	2.75 ± 0.63	2.86 ± 0.66	2.67 ± 0.60	0.08
N. remote visit/year	0.12 ± 0.42	0.14 ± 0.44	0.10 ± 0.41	0.61
Sensor experience (months)	43.2 ± 18.0	41.0 ± 18.7	45.0 ± 17.4	0.22
% of time with active sensor	89.4	93	86.58	0.13
% of time in range (70–180 mg/dL)	58.2	70.26	48.7	0.002
% di time below range (<70 mg/dL)	5.2	5.4	5.13	0.91
Mean glucose (mg/dL)	167.5 ± 35.7	146.3 ± 25.6	183 ± 34.1	<0.001
Coefficient of variation (CV) (%)	38.8 ± 7.0	36.2 ± 7.1	40.8 ± 6.4	<0.001
Associated diseases	12 celiac disease	4 celiac disease	8 celiac disease	0.52
	3 thyroid disease	2 thyroid disease	1 thyroid disease	0.39
Physical activity (hours/week)	1.61 ± 2.19	2.32 ± 2.43	1.09 ± 1.84	0.001

No.: number; T1D: type 1 diabetes; MDI: multiple daily injections; CSII: continuous subcutaneous insulin infusion; SAP: sensor-augmented therapy; HCL: hybrid closed loop; AHCL: advanced hybrid closed loop; isCGM: intermittently scanned continuous glucose monitoring (CGM); rtCGM: real-time CGM; BMI: body mass index.

**Table 2 jpm-12-01590-t002:** Family and caregiver characteristics. Mean ± SD.

	All	HbA1c ≤ 7%	HbA1c > 7%	*p* Value
No. (%)	135	57	78	
FAMILY				
Living with both parents	126 (93%)	53 (93%)	73 (94%)	0.89
Number of children	1.92 ± 0.62	1.86 ± 0.52	1.96 ± 0.69	0.35
Ethnic minority status	25 (19%)	6 (11%)	19 (24%)	0.04
Residence: urban/rural	52/83	23/34	29/49	0.67
CAREGIVER				
Mother	97 (72%)	42 (74%)	55 (70%)	0.69
Father	38 (28%)	15 (26%)	23 (30%)	0.69
Age (years)	44.3 ± 4.4	44.2 ± 4.2	44.4 ± 4.5	0.81
Marital status				
Single	5 (3%)	0	5 (6%)	-
Married	120 (89%)	52 (91%)	68 (87%)	0.46
Divorced	9 (7%)	4 (7%)	5 (6%)	0.89
Widower	1 (1%)	1 (2%)	0	-
Language currently used				
Italian	119 (88%)	50 (88%)	69 (88%)	0.35
Arabic	13 (10%)	6 (10%)	7 (9%)	0.77
Romanian	2 (1%)	1 (2%)	1 (1.5%)	0.82
Albanian	1 (1%)	0	1 (1.5%)	0.4
School education level				
Primary school diploma	3 (2%)	0	3 (4%)	0.14
Middle school diploma	35 (26%)	9 (16%)	26 (33%)	0.02
Secondary school diploma	69 (51%)	28 (49%)	41 (53%)	0.7
University studies	28 (21%)	20 (35%)	8 (10%)	<0.001
Health literacy				
Never	20 (15%)	14 (24%)	6 (7.7%)	0.006
Rarely	54 (40%)	26 (46%)	28 (36%)	0.26
Sometimes	36 (27%)	11 (19%)	25 (32%)	0.09
Often	13 (9%)	3 (5.3%)	10 (13%)	0.14
Always	12 (9%)	3 (5.3%)	9 (11%)	0.21
Working status				
Low	76 (56%)	20 (35%)	56 (72%)	<0.001
(No.unoccupied/housewife/unskilled, semi-skilled, manual workers and craftsmen)	(2/26/48)	(0/10/10)	(2/6/48)	
High	59 (44%)	37 (65%)	22 (28%)	
Nationality				
Italian	101 (75%)	45 (79%)	56 (72%)	0.35
Morocco	21 (16%)	4 (7%)	17 (22%)	0.02
Romania, Albania, Macedonia	10 (7%)	6 (11%)	4 (5%)	0.24
Others	3 (2%)	2 (3%)	1 (1%)	-
Religion				
Catholic	104 (77%)	45 (79%)	59 (76%)	0.66
Muslim	20 (15%)	5 (9%)	15 (19%)	0.09
Orthodox	10 (7%)	6 (11%)	4 (5%)	0.24
Others	1 (1%)	1 (1%)	-	-

No.: number.

**Table 3 jpm-12-01590-t003:** Scores at the Parental health locus of control questionnaire, in the two study groups. Mean ± SD.

	All	HbA1c ≤ 7%	HbA1c > 7%	*p* Value
No.	135	57	78	-
Child scale (locus 4, 9, 23, 26, 15)	4.33	4.43	4.27	0.100
Parent scale (locus 2, 13, 19, 22, 17, 24, 27)	4.50	4.64	4.39	0.001
Healthcare professionals scale (locus 5, 20, 10, 1, 3)	4.01	3.93	4.07	0.230
God scale (locus 16, 11, 21)	3.31	2.89	3.61	<0.001
Fate scale (locus 12, 18, 25, 7, 28)	2.63	2.40	2.81	<0.001
Mass media scale (locus 6, 8, 14)	2.93	2.66	3.14	<0.001

## Data Availability

The data that support the findings of this study are not publicly avail- able because they contain information that could compromise the privacy of research participants, but are available from the corresponding author upon reasonable request. Recordings and transcriptions of the semi-structured interviews are not available as they cannot be anonymised in a way that ensures the privacy or participants.

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
