# Peer review of "Influence of Parental Health Locus of Control on Behavior, Self-Management and Metabolic Control, in Pediatric Patients with Type 1 Diabetes"

_jpm, 2022, doi:10.3390/jpm12101590_

Round 1
Reviewer 1 Report
This work aimed to evaluate the effect of healthcare given by the parents on disease control of pediatric T1D. The authors collected various kinds of social economic parameters, disease related laboratory test results, patients’ baseline parameters, along with the social economic parameters of the care givers. Patients were divided into “optimal”(HbA1c ≤ 7% ) and “sub-optimal”(HbA1c > 7%) groups and the above parameters were compared between the groups. The author utilized the Wilcoxon or T-test to compare the continuous variables and calculated the spearman correlation of HbA1c and the collected parameters. This analysis revealed that patients with inferior glucose control used less insulin pump/real time CGM and were cared with an inferior social economic status. I’ve several concerns on the experimental designing, writing, and data formatting issues.
1. The analyzing procedure was not robust enough to yield reliable results. The authors only utilized the Wilcoxon or T-test to compare the continuous variables and calculated the correlation of HbA1c and the collected parameters. Usually, the parameters with p value <0.05 could be selected for logistic regression analysis. The parameters with p value <0.05 in multivariate logistic regression analysis were considered as the major contributor to the HbA1c control. Moreover, the authors calculated spearman correlation of the HbA1c (continuous variable) and the collected parameters. The authors should rather use univariate logistic regression to select significant parameters and subsequently use multi-variated logistic regression to elucidate the final contributing parameters.
2. Line 179-182: Multivariate analysis from all the cohorts revealed that mean HbA1c in the last year was positively associated with HbA1c, age, total daily insulin dose, mean glucose values, CV, and negatively with % of time with active sensor and TIR (Supplementary Material S2).
Supplementary Material S2 was the result of spearman correlation analysis not the multivariate analysis. In addition, multivariate analysis was not mentioned in Methods section.
3. Line 37, “families daily management” should be “families’ daily management”. First line indent was not set in some paragraphs (line 37, 156); line 137 seemed not to be a separate paragraph.
4. Line 170-173: Considering patients with ethnic minority status (No. 25), only 6 (24%) were at HbA1c target and only 2(8%) wear continuous subcutaneous insulin infusion (CSII) or real time continuous glucose monitoring (rtCGM) systems.
This section demonstrated the results from Table 1.B. Please insert Table 1.B in the proper place.
5. The author would better order the tables as “Table1, Table2 …” rather than “Table1.A, Table1.B…”. The captions for the abbreviations should be placed at the bottom of each table. Table 1.A calculated p value of the number of patients from two groups. What is the meaning of that? The p values would better be trimmed to the same decimal point.
Author Response
REVIEWER 1:
We would like to thank reviewer1 for her/his accurate evaluation of the study. We believe that the manuscript is significantly improved after following the comments and the suggestions of the reviewer.
- The analyzing procedure was not robust enough to yield reliable results. The authors only utilized the Wilcoxon or T-test to compare the continuous variables and calculated the correlation of HbA1c and the collected parameters. Usually, the parameters with p value <0.05 could be selected for logistic regression analysis. The parameters with p value <0.05 in multivariate logistic regression analysis were considered as the major contributor to the HbA1c control.
Moreover, the authors calculated spearman correlation of the HbA1c (continuous variable) and the collected parameters. The authors should rather use univariate logistic regression to select significant parameters and subsequently use multi-variated logistic regression to elucidate the final contributing parameters.
Thanks, linear regression analysis is reported in Supplementary material S2. Subsequently we used multiple linear regression to elucidate the final contributing parameters but we did not find any statistically significant results.
. Line 179-182: Multiple linear analysis from all the cohorts revealed that mean HbA1c in the last year was positively associated with HbA1c, age, total daily insulin dose, mean glucose values, CV, and negatively with % of time with active sensor and TIR (Supplementary Material S2). Supplementary Material S2 was the result of spearman correlation analysis not the multivariate analysis. In addition, multivariate analysis was not mentioned in Methods section.
Thanks for your suggestion. Supplementary material S2 reporting results of Spearman correlation analysis has been changed with supplementary material S2 reporting linear regression analysis results. Multivariate analysis is not mentioned in methods section because we did not perform it. It was out of the aim of the study. We apologize for the inaccuracy.
- Line 37, “families daily management” should be “families’ daily management”. First line indent was not set in some paragraphs (line 37, 156); line 137 seemed not to be a separate paragraph.
Thanks, we changed as suggested
- Line 170-173: Considering patients with ethnic minority status (No. 25), only 6 (24%) were at HbA1c target and only 2(8%) wear continuous subcutaneous insulin infusion (CSII) or real time continuous glucose monitoring (rtCGM) systems. This section demonstrated the results from Table 1.B. Please insert Table 1.B in the proper place.
Thanks, we moved this sentence to section 3.1.2, where data reported in Table 1B (renamed Table 2) are described.
- The author would better order the tables as “Table1, Table2 …” rather than “Table1.A, Table1.B…”. The captions for the abbreviations should be placed at the bottom of each table. Table 1.A calculated p value of the number of patients from two groups. What is the meaning of that? The p values would better be trimmed to the same decimal point.
Thanks, we changed as suggested
Reviewer 2 Report
In this paper, the authors aimed to evaluate the Parental Health Locus of Control (PHLOC) in the primary caregiver of youths with T1D and its relationship to the diabetes behavior, self-management and metabolic control.
The paper is interesting, but I have two concerns that must be addressed by authors.
Study design
The authors indicated as “optimal glycemic group” that one with HbA1c<7%. In my opinion, it is necessary to refer to the absence of hypoglycemic events.
Results and Discussion
The authors refer to “ethnic minority groups”. Can the authors exclude that this is a surrogate of the poverty? As it is well known, the socioeconomic status is a major determinant of disease and its prognosis, even in a country with a public Health Care System, such as Italy.
Author Response
REVIEWER 2:
We would like to thank reviewer2 for her/his accurate evaluation of the study. We believe that the manuscript is significantly improved after following the comments and the suggestions of the reviewer.
The paper is interesting, but I have two concerns that must be addressed by authors.
- Study design. The authors indicated as “optimal glycemic group” that one with HbA1c<7%. In my opinion, it is necessary to refer to the absence of hypoglycemic events.
Thanks, we added that we used HbA1c to classify patients in “optimal control” as suggested by ISPAD for epidemiological studies [1] and as reported in previous studies on HLOC [5,6].
We added as a limit that we could not use time below range (TBR) < 5% as an inclusion criteria, because our patients wore different types of sensors and data were not comparable.
Only one patient in the “sub-optimal group” had severe hypoglycemia, as reported in the results.
- Results and Discussion: The authors refer to “ethnic minority groups”. Can the authors exclude that this is a surrogate of the poverty? As it is well known, the socioeconomic status is a major determinant of disease and its prognosis, even in a country with a public Health Care System, such as Italy
Thanks, we evaluated SES collecting data regarding parents’ educational and occupational situation, and we reported in ethnic minority groups reduced school education and health literacy levels. We did not have reliable data on economic income and we added this as a limit.
Round 2
Reviewer 1 Report
The revised manuscript has addressed the major comments of the original version.
Reviewer 2 Report
The authors addressed all my concerns.